# The Use of Virtual-Problem-Based Learning during COVID-19 Pandemic Emergency: Veterinary Students’ Perception

**DOI:** 10.3390/vetsci9100581

**Published:** 2022-10-21

**Authors:** Fabiana Micieli, Giovanni Della Valle, Chiara Del Prete, Paolo Ciaramella, Jacopo Guccione

**Affiliations:** Department of Veterinary Medicine and Animal Production, University of Naples Federico II, 80137 Naples, Italy

**Keywords:** veterinary clinically training, student survey, teaching method

## Abstract

**Simple Summary:**

New teaching methods become more popular during the COVID-19 pandemic emergency to guarantee high educational standards despite the strict rules of social distancing and lockdown. Problem-based learning represents one of the most popular and flexible teaching systems, replicating real-life experiences and stimulating the integration of knowledge and lifelong learning skills. The authors aimed to assess student satisfaction with virtual-problem-based learning compared to the “traditional and in presence” clinical training.

**Abstract:**

The COVID-19 global pandemic emergency forced us to replace the “traditional and in presence” clinical, pre-graduating, veterinary medical training with clinical virtual-problem-based learning (v-PBL). This prospective cross-sectional case-control study aimed to evaluate the students’ perception of the v-PBLs compared to the traditional veterinary clinical training (t-VCT). The t-VCT consisted of supervised management of clinical cases admitted at the Veterinary Teaching Hospital and in the field. The v-PBL consisted of genuine clinical cases shared by tutors throughout an online platform. A survey was delivered to all the fifth-year students who completed the t-VCT or the v-PBL. The survey was completed by 49% of the students. Overall student satisfaction regarding the training experiences was high in both groups, but it was less in the v-PBL than in the t-VCT group. The students of the v-PBL group perceived that they could not improve their practical clinical skills through online sessions, and they emphasized how it could be employed as support for traditional practical activities. All the students are satisfied with the supervision and considered the training correctly focused on relevant learning objectives and the task clearly explained. Stimulating the integration of knowledge and lifelong learning skills replicating life experiences the v-PBLs represented an attractive curricular alternative for veterinary education.

## 1. Introduction

The pandemic emergency due to the coronavirus disease-2019 (COVID-19) is forcing veterinary colleges all over the world to modify deeply the traditional teaching approach employed. Despite the strict rules of social distancing and lockdown applied during the pandemic in all countries, veterinary colleges have to guarantee the achievement of all goals set in their education core *curriculum*. High educational standards are suggested for veterinary students by international organizations such as “Office International des Épizooties” or “European Association of Establishments for Veterinary Education”. Now more than ever, the goal can be reached by providing high-quality education through the use of premium quality learning resources and technologies for continuing to meet students’ expectations and to guarantee an exciting educational process [1,2]. This period of uncertainty is forcing a significant revision of fundamentals in veterinary education through a deep self-evaluation of the teaching methods used, so far. If parts of a constructive alignment such as teaching objective, assessment approaches, and assessment criteria may be only partially involved, the same does not seem for the teaching methods and learning outcomes [1,2]. Indeed, the restrictive rules adopted by the different countries forces the Academia to replace traditional face-to-face lectures with innovative teaching methods (e.g., virtual classes, tutorial, simulations, distance learning, web-based learning, etc.).

In veterinary education, problem-based learning (PBL) represents one of the most popular and flexible teaching systems, cultivating students’ self-directed abilities [3]. It replicated real-life experiences stimulating the integration of knowledge and lifelong learning skills commonly required for a veterinary medical education [3]. The PBL process has been described as “fun”, “interesting”, “motivating”, “stimulating”, and “engaging” by the students [4]. Its success depends on (i) the careful selection of genuine clinical cases; (ii) the possibility of integration of prior knowledge gained by the students through the entire *curriculum*; (iii) the ability to improve critical thinking, problem-solving abilities, independent learning, and a holistic approach to a case [5,6]. For a long time, the PBL has been cited as an attractive curricular alternative for veterinary education, prompting several Veterinary Colleges worldwide to integrate permanently the system within their curricula [1,7,8,9]. Despite the cited advantages, some studies questioned the ability of PBL to improve students ‘core knowledge’ [10]. Other authors complained about this method because the teaching staff conceives it as time-consuming and labor-intensive [11].

Due to the complete lookdown adopted by the Italian Government during spring 2020 and 2021, the Department of Veterinary Medicine and Animal Production of Napoli (Naples, Italy), decided to replace the “traditional and in presence” hours (h) dedicated to the clinical, pre-graduating, veterinary medical training with multiple sessions of clinical virtual-problem-based learning (v-PBL).

This study aimed to verify whether v-PBLs replicating life experiences could be considered a useful alternative to the traditionally employed didactic form of hands-on learning through a students’ feedback survey analysis.

## 2. Materials and Methods

### 2.1. Study Design and Sample Population

The present retrospective, cross-sectional, case-control study was based on the results of a students’ survey questionnaire. The latter was performed to evaluate the pre-graduate students’ perception of the clinical v-PBLs compared to the traditional veterinary clinical training (t-VCT). The study population was represented by all the fifth-year veterinary students who completed the t-VCT (76 students) or the clinical v-PBL (46 students). Students that performed the t-VCT (between March 2019 and September 2019) were enrolled as Group Traditional (Group T), while those that attended clinical v-PBL (between March 2020 and September 2020) as Group Virtual (Group V). All of them received the online, voluntary, satisfaction questionnaire by e-mail at the end of their training program. The sample population was made up of students deciding to participate.

### 2.2. Practical Training Characteristics

Students belonging to Group T performed a standard t-VCT fully approved by the EAEVE and achieved the requirements of the standard UNI EN ISO 9001-2015 (n. 317didaSGQ00). The t-VCT consisted of 480 h of clinical rotation divided into 240 h in small animals (i.e., dog, cat, unconventional animals) and 240 h in large animals (i.e., buffalo, cow, horse, sheep/goat, and pig). The student in Group V performed a clinical v-PBL made up of 240 h divided into 120 h dedicated to the small animal practice and 120 h to large animal practice. Surgery, anesthesia, emergency, intensive care, internal medicine, dermatology, reproduction, neonatology, and imaging were the possible application fields for both training and in all the species considered.

Students in Group T mainly managed clinical cases admitted at the Veterinary Teaching Hospital and referred ones (directly in the field). All the activities were supervised by tutors (academic staff) with a maximum tutor: students ratio of 1:5. In Group V training consisted of genuine clinical cases shared by tutors on the Microsoft Teams^®^ platform (Microsoft Corporation, Redmond, DC, USA) with a maximum tutor: students ratio of 1:8.

An overall amount of n 18 virtual clinical cases were performed by each student. Thirteen hours was the planned hourly commitment/case. The time was approximately divided as follows: 1 h of sharing activity in a virtual room (i.e., tutors briefly introduced the case, and gave useful support and teaching materials to manage it); 8 h of self-learning (team-working allowed) and problem-solving activities (individual-work recommended); 4 h of virtual Group discussion activity (i.e., either between students’ tutor directed or between tutor and students). All the v-PBLs were based on clinical cases originating from tutors’ real-life experience and structured according to a constructive alignment including activities of evaluation (e.g., judges, criticizes, compares, etc.); synthesis (e.g., combines, creates, constructs, etc.); analysis (e.g., differentiates, estimates, diagrams, etc.); application (e.g., demonstrate, solves, modifies, etc.); comprehension (e.g., classifies, explain, predicts, etc.) and knowledge (e.g., identifies, describes, matches, etc.) [12]. Overall, each clinical case consisted of 15 to 20 progressive questions (e.g., closed-ended questions, long/short open-ended questions, etc.) and activities related (e.g., comments and/or making figures, tables, graphs, concept maps, etc.).

### 2.3. Survey Questionnaire

The questionnaire was created on the Microsoft Forms^®^ platform (Microsoft Corporation, Redmond, DC, USA).

The questionnaire was first administered to a representative sample of students (intended respondents) for initial validation. Based on the global averages by day regarding the highest click-through rates for surveys [13], the questionnaire was sent on Monday morning. Students accepted to participate in the anonymous questionnaire were previously informed about the potential use of the data for scientific purposes; by submitting a questionnaire reply, they accepted the aims and methods used. To further validate the survey, the authors reviewed answers from the initial ten questionnaires, and since students’ answers were as anticipated, the questions were not changed.

The survey was made up of 18 items, divided into five sections: “demographic data”, “satisfaction”, “clinical skills”, “supervision” and “student’s perceptions of the training”. Demographic questions, including age and sex, were used to characterize the study sample. The “satisfaction” section included: overall students’ satisfaction, the organizational aspects of the training, and students’ expectation about clinical training. Furthermore, the questionnaire investigated the student’s perception of the knowledge gained with the clinical training (“clinical skills” section) and of the clinical supervision (“supervision” section). The section entitled “student’s perceptions of the training” investigated difficulties encountered during the training. The contents of the questionnaire are reported in detail in Table A1/Table 1. The best practices for customer survey satisfaction were followed to make both questions and answers [14]. Briefly, questions were structured as clear, concise, and fulfilling the end goal. They were asked once a time, avoiding leading and loading questions, getting specific during formulation, and avoiding assumptions. The types of answers were: consistent Likert-scale (containing 5 response options), multiple choice, and open-ended questions.

### 2.4. Data Management and Statistical Analyses

Data were expressed as absolute numbers, percentages, medians, and interquartile ranges (Q1, Q3) and were analyzed by standard descriptive statistics. The normality has been assessed by means of Shapiro-Wilk tests, normal probability plots, and histograms.

The Likert questions (L-Qs, Table A1) have been numbered progressively throughout the different survey’s sections: L-Qs from 1 to 6 (Section-2-satisfaction), L-Qs from 7 to 10 (Section-3-clinical skills), L-Qs 11 to 13 (Section-4-clinical supervision) and L-Q 14 (Section-5-perceptions of the problem). For Section 3, the abilities reached have been quantified using a delta (Δ) between the mean scores given at the beginning of the training and those at the end. This variable has been hereby defined as Δ-answer. Moreover, for all the L-Qs the possible answer options have been scored into five progressive points categories (1 “strongly disagree/Very poor/Too easy” to 5 “strongly agree/Very good/Too difficult”), except for L-Q6 including only 4 possible progressive answers scored into as many points-categories (2 “yes” to 5 “No, it was too long”).

The reliability of the internal consistency of the questionnaire was tested using Cronbach’s alpha coefficient, producing acceptable values for alpha, ranging from 0.70 to 0.95 [15]. All statistical data were analyzed using dedicated software (SPSS, Version 26.0.0, Chicago, IL, USA).

#### 2.4.1. Inter-Group Comparison

For each L-Q, the median value of the answers’ scores was also calculated and inter-groups differences were assessed by the Mann–Whitney U-test for two independent samples. Probabilities <0.05 were considered statistically significant for all tests used.

For each L-Q (Δ-answer’s results included), the inter-group comparison was performed assessing the differences between expected and observed response frequencies, using χ^2^-test with two ways contingency tables. An inter-group difference between the response frequencies for each of the available options was assumed as H0. Even in this case, the latter was rejected by significant values, demonstrating the presence of a difference between the available options. Fisher’s exact test was performed in case of low expected frequencies (<5).

#### 2.4.2. Intra-Group Comparison

For all L-Qs, the response frequencies for the available options have been calculated. The non-parametric χ^2^-test was used to assess the distribution of the response frequencies to the individual L-Qs, assuming them as uniform (null hypothesis-H0). Significant values rejected the H0, showing a lack of choice consistency between the available options.

#### 2.4.3. Open-Ended Questions

The answers to the two open-ended questions (OE-Q) belonging to Section 2 (OE-Q1) and Section 5 (OE-Q2), were simultaneously read by the authors, and a majority-vote system was used to assess if answers stuck to the question. All those not focused on the OE-Qs’ topic were discarded, while those accepted were retrospectively classified into one or more categories.

## 3. Results

Overall, the survey was sent to 122 students and 60 (49.1%) of them completed the questionnaire (36.7% male and 63.3% female). The different ages were distributed as follows: 35.0% range between 22 and 24 years, 51.7% between 25 and 27, 8.3% between 28 and 30, and 5.0% belonged to the category >30. Twenty-eight students have been enrolled in Group T and 32 in Group V. Median values observed for each section are shown in Figure 1. Cronbach’s alpha coefficient for internal consistency for the questionnaire was 0.9.

### 3.1. Inter-Group Comparison

The comparison between the recorded median scores between groups is shown in detail in Table 1. Briefly, statistical differences were found for L-Q4 (*p* = 0.001) and L-Q7 (*p* = 0.034).

The inter-group comparison between the response frequency for each of the available options, showed in Section-2 a difference for L-Qs 1-value 3 (*p* = 0.226), L-Qs 3-value 2 and 5 (*p* = 0.286 and *p* = 0.250), L-Qs 4-value 1 (*p* = 0.143), L-Qs 5-value 3 and L-Qs 6-value 4 (*p* = 0.056). In Section-3 analogous results were observed for L-Qs 7-value 2 (*p* = 0.214), while in Section 4 for L-Qs 11-value 2 (*p* = 0.107) and L-Qs 13-value 2 (*p* = 0.132). The H0 was rejected for all the remaining answers in both groups (Δ-answer included).

### 3.2. Intra-Group Comparison

The distribution of the response frequencies to the individual L-Qs (Table 1).

Group T showed a choice consistency for L-Qs 4-7-8-10, while Group V for L-Qs 5-6. The H0 was rejected for all the remaining answers in both groups. For the quantification of skills reached by Δ-answer, in Group T, 50.0% of the students indicated an improvement of one point, 3.6% of 2 points, while the remaining 46.4% did not report any difference. In Group V, 37.5% of the students indicated an improvement of one point, the 3.1% of 2 points, while the remaining 59.4% did not report any difference.

### 3.3. Open-Ended Questions

Regarding the OE-Q1, 60.7% of the answers in Group T, and 43.7% in Group V stuck to the question. All of them were classified as “expectation of acquiring new skills”. None of OE-Q2′s answer was instead discharged. They were classified as “no suggestions” [10.7% for Group T—56.3% for Group V], “suggestions regarding the format of the training performed” [53.6% for Group T—43.8% for Group V], “suggestions to improve their clinical skills” [35.7% for Group T—0% for Group V], and “suggestions to improve the interaction with tutors“ [21.4% for Group T—0% for Group V].

All students of Group V had no suggestions to improve the v-PBL and emphasized the use of this training in addition to the traditional one in the future.

## 4. Discussion

The current study aimed to assess the students’ perception of v-PBLs compared to the t-VCT by a well-defined custom-model survey.

Nowadays, almost all Universities routinely evaluate students’ feedback regarding the teaching strategy to improve the quality of the offered curricula. Students’ answers can be used as a reasonably reliable and valid information source for the effectiveness of the learning experience [16,17,18]. Nevertheless, the necessity to introduce alternative teaching systems during such an uncertain period forced the authors to a careful and methodical approach to the outcomes evaluation. To objectively evaluate the findings observed, the current questionnaire has been conceived following the guidelines for customer survey satisfaction [13,14]. Even if the survey has been developed carefully, an overall high proportion of non-responder students has been found (50.9%). For a long time, the challenge of nonresponse is considered an unresolved issue in survey research [19,20,21]. Different studies investigated the reasons influencing the students’ decision to participate in the surveys [22,23,24]. Beyond such factors as personality and educational level, it has been revealed how the gender and age of the participants may significantly influence the results considering that female and younger students would seem to be the most likely participants [22]. Accordingly, in the present study, most of the responders were female (63.3%) and with an age range between 22 and 24 years (35%), although this data could be affected by the veterinary student population.

The PBL is worldwide used in many medical schools having a relevant impact on the student’s education [3]. It does not just represent problem-solving per se, but it uses the appropriate issues as an exercise in clinical reasoning to improve knowledge and understanding [3]. In our setting students in Group V felt the experience was less satisfying, as compared to Group T, but both considered the training correctly focused on relevant learning objectives with tasks clearly explained. Students receiving v-PBL revealed a better distribution of the answers towards positive judgments for L-Q2 and L-Q3 (Table 1). These results could be explained considering that Group V performing a remote activity, had time to be guided by a supervisor, acquire the information, argue about the issue, and look for a solution directly at home. On the contrary, students belonging to Group T, following the daily frenetic clinical practice, could be more stressed and anxious in managing real, self-governed, situations.

As reported in the literature, although the PBL process has been described as interesting and motivating by students, the perception of its didactic effectiveness could be negatively influenced by several factors [19]. In the present study, looking at L-Q4′s answers, a considerable percentage of the students belonging to Group V was indifferent to the teaching system (25.0%—neither agree nor disagree) or disappointed (3.1% strongly disagree—15.6% disagree). A too-fast transition from traditional teaching methods to new ones (unexpected use of different teaching systems could produce anxiety), inappropriate division in PBL groups according to their cultural background (tutors should be aware of the cultural differences during the groups’ establishment), as well presence dysfunctional groups (e.g., too big, lateness, absenteeism, students too quiet or dominant) are considered as a negative factor influencing student’s perception [11,25,26].

Regarding L-Q5 (Table 1), more than 50% of the students belonging to Group T believed the clinical training met their expectations (46.4% agree—14.3% strongly agree), while Group V did not express a clear opinion regarding this point. These results could be explained by considering the organization of the practical activities along their *curriculum*, thought to reach the One-day competencies recommended by EAEVE. Indeed, students in our department start doing practical activities in the first year (with animal handling), arriving at the final one able to perform self-governed clinical procedures. During the *curriculum*, our students significantly improve their competencies and skills, but only during their final one they can try their hand at attending the in-presence-clinical-training with a high grade of fulfillment. According to the results of our survey, despite all the students arriving at the final years with the same clinical skills, only the students of Group T showed higher satisfaction with the clinical skills acquired during the training. Finally, a difference concerning answers distribution was found also for L-Q6 (Table 1). Group T felt as correct the duration of the clinical activities performed, while Group V did not express a clear opinion. Group T is probably perceived to have finalized all the efforts performed during the entire *curriculum*, acquiring awareness of the abilities and verifying their competencies in real self-governed activities; on the contrary, Group V could not put himself to the test. The time spent managing cases replicating life experiences at home could be perceived as less productive and not enough as compared to activities in person.

Veterinary graduates need to be effective and efficient in clinical work; therefore, learning clinical skills is a key component. This assumption makes interesting the interpretation of the answers belonging to section-3-Clinical Skills. Both the groups showed an overall improvement in their skills (Δ-answer) and felt to have the chance to deepen their knowledge (L-Q9) but, on the other hand, they fill not confident about taking autonomous clinical decisions (L-Q10). For this specific answer, have to be highlighted how for Group T this could be a predictable opinion because managing complex case self-governed require time and goes far beyond EAEVE recommendations for graduating students. On the contrary, Group V did not test itself with real self-governed and did not develop the ability to take clinical decisions having experienced only clinical cases replicating real life and directing the student towards clinical pre-defined deductions [1].

Our findings are consistent with results from similar studies conducted on the perceptions of undergraduate medical students toward fully online training [27,28]. Virtual PBL cannot completely replace traditional clinical training, practical sessions during veterinary school are required to allow the students to learn the skills needed for their careers [3,10,29]. Nevertheless, improvement of linguistic skills, self-confidence, and reasoning was recently proved with the use of v-PBL in clinical contexts [30]. Several studies reported that students felt their knowledge improved after the integration of PBL into a traditional veterinary *curriculum* and do not suppose an exclusive use of this teaching system [3,9,30,31,32].

Teaching methods are considered successful only when a deep synergy between academic staff and students is established [3,19]. This relationship plays an important role in the learning process in medical education [33]. A balance between closeness and the necessary distance in the student-teacher relationship is required to obtain the optimal learning experience and the professional development of the students [34]. Despite the social distancing imposed by the COVID-19 pandemic, the students were overall satisfied with the tutor’s supervision received in both groups recognizing them as a supporting (L-Q11) and encouraging role (L-Q12) that met their expectations (L-Q13). This consistency of the result between groups is probably due to the use of synchronous e-learning during the v-PBL, contributing to improving Group V’s opinion. Synchronous e-learning is closer to face-to-face communication due to the real-time, active interaction between student and teacher and is reported to increase motivation and physiological arousal in students [35]. In a recent study on perceptions of e-learning during the COVID-19 pandemic, students from different countries revealed contrasting opinions regarding the ease of interaction with lecturers and peers during the online sections [36]. The survey was not conclusive because the population divided itself between those for and against it [36]. The results of the present study underling as the interaction between staff and students have to be steady to guarantee positive learning outcomes. The assumption seems to be confirmed by OE-Q2 where no suggestion to improve the interaction with tutors has been recorded for Group V, probably because already been deemed satisfactory. Teaching staff not committing to the process could be a critical point. Changes in teaching strategies without consultation, inadequate training, and the additional time required for the teaching staff may produce a lack of commitment and goodwill, reducing the impact of the virtual training [19].

Finally, the Authors would like to make some considerations about the questionnaire used and the type of answers received. Based on the usefulness of the information obtained, it may represent a model for further analysis of medical students’ perceptions undergoing clinical-practical training (regardless of the in-person or virtual mode). As previously explained, some of the most recent guidelines for customer survey satisfaction were followed to build up and deliver it to the students [13,14]. Nevertheless, data analysis and interpretation should always take into consideration the human emotional component that can influence the answers given and that no questionnaire/survey can exclude. The Authors tried to quantify as much as possible the influence of this variable employing Cronbach’s alpha coefficient, but they could not avoid considering the emotional component during the explanation of the results. It cannot be excluded that the use of other systems of accuracy evaluation might improve the quality of the information obtained and the performance of the questionnaire, even more [37].

## 5. Conclusions

Due to the current COVID-19 pandemic emergency, the Department of Veterinary Medicine and Animal Production of Napoli (Italy) decided to replace its well-established-EAEVE approved-didactic approach, based on “in presence”—clinical—pre-graduating—veterinary medical training, with multiple sessions of clinical v-PBL. According to our results, overall, Group T is probably perceived to have finalized all the efforts performed during the entire *curriculum*, acquiring awareness of the abilities, and verifying their competencies in real self-governed activities. On the other side, Group V could not put himself to the test and was perceived as less productive and not enough as compared to activities in-person. If on the one hand, the v-PBL may show some limitations negatively influencing students’ perception, on the other hand, it can be considered a good option during the current period of social distancing, to allow the students to complete their mandatory clinical practical training. Looking at the analysis of the responses, the traditional in-presence practical activities cannot be replaced in normal condition because it represents an essential base for a core *curriculum* with international standard. Nevertheless, the use of well-organized v-PBL sections is advisable to integrate the traditional practical teaching approach and to improve students’ clinical reasoning and motivation.

Given the importance of the implementation of the future veterinary *curriculum* with virtual teaching systems, further studies are needed to explore drawbacks and to find the correct balance between virtual and in-presence activities. Moreover, the evaluation of practical “hands-on” skills of students exposed to virtual teaching should be done to promote the use of alternative teaching systems on a regular basis.

## Figures and Tables

**Figure 1 vetsci-09-00581-f001:**
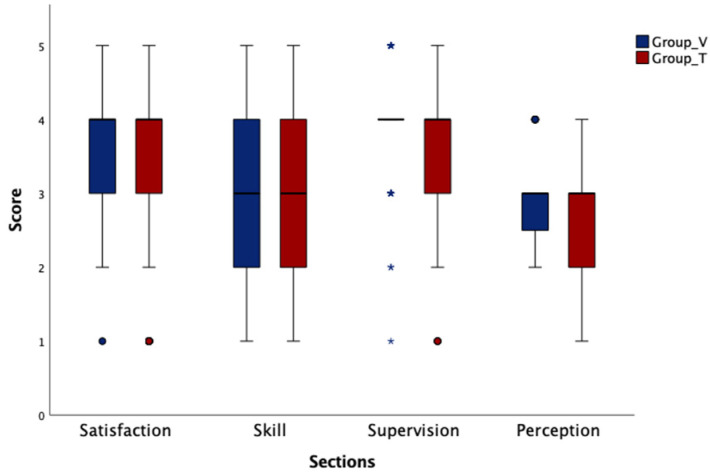
Median values observed for each section in Group Traditional (Group_T) and Group Virtual (Group_V). For each box, the central line represents the median, the edges of the boxes represent the IQR (25th and 75th percentiles), and the whiskers represent the extreme points. ● Single data point outliers * multiple data point outliers (The size of the asterisk is proportional to the quantity of outliers).

**Table 1 vetsci-09-00581-t001:** Answers to the Likert questions. Statistical analysis and the number of participants for Group Traditional (T) and Group Virtual (V) are reported.

Likert Question (L-Q)	Question	Group	Response Value					H0	Median (Q1, Q3)	*p*
			1	2	3	4	5			
	Satisfaction									
L-Q1	On a scale of 1 to 5 (with 1 being least satisfied and 5 being most), please rate your overall satisfaction with clinical training.	T	2 (7.1%)	3 (10.7%)	3 (10.7%)	15 (53.6%)	3 (10.7%)	0.004	3 (3, 4)	0.100
		V	0 (0%)	4 (12.5%)	15 (46.9%)	10 (31.2%)	3 (9.4%)	0.008	4 (3, 4)	
L-Q2	The clinical training was focused on relevant learning objectives.	T	0 (0%)	4 (14.3%)	2 (7.1%)	18 (64.3%)	4 (14.3%)	0.000	4 (4, 4)	0.361
		V	0 (0%)	2 (6.2%)	3 (9.4%)	20 (62.5%)	7 (21.9%)	0.000	4 (4, 4)	
L-Q3	My tasks during clinical training were clearly explained.	T	2 (7.1%)	6 (21.4%)	3 (10.7%)	16 (57.1%)	1 (3.6%)	0.000	4 (4, 5)	0.710
		V	0 (0%)	1 (3.1%)	2 (6.2%)	22 (68.7%)	7 (21.9%)	0.000	5 (4, 5)	
L-Q4	The organization of clinical training was appropriate.	T	4 (14.3%)	6 (21.4%)	5 (17.8%)	13 (46.4%)	0 (0%)	0.067	4 (4, 4)	0.001
		V	1 (3.1%)	5 (15.6%)	8 (25%)	15 (46.9%)	3 (9.4%)	0.001	4 (2, 4)	
L-Q5	The clinical training meet my expectations.	T	1 (3.6%)	8 (28.5%)	2 (7.1%)	13 (46.4%)	4 (14.3%)	0.002	4 (3, 4)	0.132
		V	0 (0%)	10 (31.2%)	7 (21.9%)	11 (34.4%)	4 (12.5%)	0.290	3 (2, 4)	
L-Q6	Was the duration of clinical training sufficient to ensure adequate competencies?	T	-	4 (14.3%)	6 (21.4%)	2 (7.1%)	16 (57.1%)	0.001	3 (2, 4)	0.623
		V	-	0 (0%)	10 (31.2%)	7 (21.9%)	15 (46.9%)	0.216	4 (2, 4)	
	Clinical Skills									
L-Q7	How would you define your clinical experience before this clinical training?	T	0 (0%)	1 (3.6%)	9 (32.1%)	11 (39.3%)	7 (25%)	0.046	3 (2, 3)	0.034
		V	0 (0%)	6 (18.7%)	13 (40.6%)	9 (28.1%)	4 (12.5%)	0.124	2 (1.7, 3)	
L-Q8	How would you define your clinical experience after this clinical training?	T	0 (0%)	7 (25%)	8 (28.5%)	9 (32.1%)	4 (14.3%)	0.572	3 (2, 4)	0.158
		V	1 (3.1%)	14 (43.7%)	7 (21.9%)	7 (21.9%)	3 (9.4%)	0.000	3 (2, 3.2)	
L-Q9	I had the chance to deepen my knowledge through this clinical training.	T	2 (7.1%)	1 (3.6%)	4 (14.3%)	19 (67.8%)	2 (7.1%)	0.000	4 (4, 4)	0.124
		V	0 (0%)	1 (3.1%)	3 (9.4%)	23 (71.9%)	5 (15.6%)	0.000	4 (3.7, 4)	
L-Q10	During this clinical training, I attained the ability to make autonomous clinical decisions.	T	4 (14.3%)	8 (28.5%)	11 (39.3%)	5 (17.8%)	0 (0%)	0.232	3 (2, 4)	0.193
		V	0 (0%)	13 (40.6%)	8 (25%)	8 (25%)	3 (9.4%)	0.013	3 (2, 3)	
	Clinical supervision									
L-Q11	During this clinical training, supervisors supported in managing clinical cases.	T	1 (3.6%)	3 (10.7%)	3 (10.7%)	17 (60.7%)	4 (14.3%)	0.000	4 (4, 4.2)	0.213
		V	1 (3.1%)	1 (3.1%)	3 (9.4%)	19 (59.4%)	8 (25%)	0.000	4 (3.7, 4)	
L-Q12	Supervisors encouraged students to deepen their knowledge of issues encountered during this clinical training.	T	1 (3.6%)	3 (10.7%)	6 (21.4%)	12 (42.8%)	6 (21.4%)	0.002	4 (4, 4)	0.233
		V	0 (0%)	0 (0%)	5 (15.6%)	21 (65.6%)	6 (18.7%)	0.001	4 (3, 4)	
L-Q13	During this clinical training, the supervisors’ support meets my expectations.	T	1 (3.6%)	6 (21.4%)	3 (10.7%)	13 (46.4%)	5 (17.8%)	0.002	4 (3.7, 4.2)	0.185
		V	0 (0%)	1 (3.1%)	7 (21.9%)	16 (50%)	8 (25%)	0.003	4 (2.7, 4)	
	Student’s perceptions of the training									
L-Q14	How would you define the clinical activities of this training?	T	2 (7.1%)	9 (32.1%)	15 (53.6%)	2 (7.1%)	0 (0%)	0.005	3 (2.7, 3)	0.132
		V	0 (0%)	8 (25%)	19 (59.4%)	5 (15.6%)	0 (0%)	0.006	3 (2, 3)	

## Data Availability

Not applicable.

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
