# Peer review of "The Use of Virtual-Problem-Based Learning during COVID-19 Pandemic Emergency: Veterinary Students’ Perception"

_vetsci, 2022, doi:10.3390/vetsci9100581_

Round 1

Reviewer 1 Report

Research at this article is very interesting and especially important for all teachers who met with online teaching during the pandemic.

Authors made very good questionnaire with five sections.

Specially is very good that they had different groups of students and that they were older students who could make comparisons with the period before and during the pandemic.

The data analysis is very detailed with good tables.

Some of the results should be shown on graphs for better visibility.

The conclusion is very short and needs to be expanded.

If it is possible to compare the obtained results with similar research in the world (even if they are not veterinary students, but related fields).

In this way, the list of references would be more complete with more recent articles.

Author Response

REVIEWER 1

Research at this article is very interesting and especially important for all teachers who met with online teaching during the pandemic.

Authors made very good questionnaire with five sections.

Specially is very good that they had different groups of students and that they were older students who could make comparisons with the period before and during the pandemic.

The data analysis is very detailed with good tables.

AU: Thank you very much for your comments. For the Authors believe that new and alternative teaching systems are the future to improve the quality of the core curriculum of veterinary students.

Some of the results should be shown on graphs for better visibility.

AU: Thank you for your suggestion. The text has been modified accordingly. Please, see the text (Results Section)

The conclusion is very short and needs to be expanded.

AU: Thank you for your suggestion. The text has been modified accordingly. Please, see the text (Conclusions Section)

If it is possible to compare the obtained results with similar research in the world (even if they are not veterinary students, but related fields).

AU: Thank you for your suggestion. The text has been modified accordingly. Please, see the text (Discussion Section)

In this way, the list of references would be more complete with more recent articles

AU: Thank you for your suggestion. The text has been modified accordingly. Please, see the text (Reference Section)

Reviewer 2 Report

 I found that the number of responding students was low to reach a conclusion. However, it is a subject that deserves to be known by veterinary schools because it can generate similar research in other universities.

I have only few suggestions in the attached manuscript file.

Author Response

REVIEWER 2

I have only few suggestions in the attached manuscript file.

AU: Thank you for your suggestion. The text has been modified accordingly. Please, see the text (Introduction Section)

Reviewer 3 Report

The manuscript at hand evaluates virtual against “in person” training of veterinary medicine students using a specifically designed questionnaire. They find that virtual teaching is an adequate method to replace “in person” training to a certain degree, but cannot replace it fully. Given the changes to studying, teaching and learning routines brought on by the Covid-19 pandemic this is a very timely topic and the findings are helpful in regards of designing teaching curriculums in the future. While this manuscript deals with students of veterinary medicine specifically and only represents a case study, the findings can be insightful for other instiutions, as well as for other fields of study, where decisions have to be made regarding the use of virtual versus “in person” teaching methods.

 --------------------------------------------------------------------------

Comments on the manuscript

The manuscript is in need for an edit by an English native speaker. There are issues with grammar, typos and translations, which sometimes diminishes clarity of what the authors are trying to say.

I found the result section hard to follow in places. This might change with an edit of the language; nevertheless I think for the sake of structure and clarity it would be beneficial if the authors clearly indicate what they consider the key findings/key take home message of this survey. 

In regards to data presentation, I would find it helpful if some of the data could be visualized in plots since it can get a bit tedious to go back and forth between text and table constantly. For example, it might be useful to visualize the data that supports the key findings of this study.
The visualization is not a must, but I would find it helpful.

Given the current importance of the topic and the need to consider virtual teaching for future curriculums, I would find it very helpful if the authors extended the discussion of their conclusions to include suggestions/thoughts on implementing virtual teaching (e.g. drawbacks, examples, balance virtual VS. "in person"), as well as perhaps giving an outlook into future evaluations/studies regarding this topic (e.g. evaluating practical “hands on” skills of students after virtual teaching).

It might be an interesting bonus to include some examples answers from the OE-Q1 & Q2 in the appendix. But again, not a must.

Author Response

REVIEWER 3

The manuscript is in need for an edit by an English native speaker. There are issues with grammar, typos and translations, which sometimes diminishes clarity of what the authors are trying to say.

AU: Thank you for your suggestion. The text has been proofread to improve its quality. Please, see the text

I found the result section hard to follow in places. This might change with an edit of the language; nevertheless I think for the sake of structure and clarity it would be beneficial if the authors clearly indicate what they consider the key findings/key take home message of this survey. 

AU: Thank you for your suggestion. The text has been modified accordingly. Please, see the text (Results Section)

In regards to data presentation, I would find it helpful if some of the data could be visualized in plots since it can get a bit tedious to go back and forth between text and table constantly. For example, it might be useful to visualize the data that supports the key findings of this study.
The visualization is not a must, but I would find it helpful.

AU: Thank you for your suggestion. The text has been modified accordingly. Table has been revised and a box plot has been added.  Please, see the text (Results Section)

Given the current importance of the topic and the need to consider virtual teaching for future curriculums, I would find it very helpful if the authors extended the discussion of their conclusions to include suggestions/thoughts on implementing virtual teaching (e.g. drawbacks, examples, balance virtual VS. "in person"), as well as perhaps giving an outlook into future evaluations/studies regarding this topic (e.g. evaluating practical “hands on” skills of students after virtual teaching).

AU: Thank you for your suggestion. The text has been modified accordingly. Please, see the text (Conclusions Section)

It might be an interesting bonus to include some examples answers from the OE-Q1 & Q2 in the appendix. But again, not a must

AU: Thank you very much for your suggestion. A classification of the answers has been already proposed by the authors in results section in order to give an idea to the readers. In our opinion the inclusion of a single, selected, raw answer could not fulfil all the possible scenarios detected.
